

# Simultaneous state-parameter estimation of rainfall-induced landslide displacement using data assimilation

Jing Wang[1], Guigen Nie[1,2], Shengjun Gao[3],Changhu Xue[1]

[1] GNSS Research Center, Wuhan University, Wuhan, 430079, China

[2] Collaborative Innovation Center for Geospatial Information Technology, Wuhan, 430079,China

[3] Chinese Antarctic Center of Surveying and Mapping, Wuhan, 430079, China

*Corresponding to*: Guigen Nie (ggnie@whu.edu.cn)

**Abstract**. Landslide displacement prediction has great practical engineering significance to landslide stability evaluation and early warning. The evolution of landslide is a complex dynamic process, applying classical prediction method will result in significant error. Data assimilation method offers a new way to merge multi-source data with the model. However, data assimilation is still deficient in the ability to meet the demand of dynamic landslide system. In this paper, simultaneous state-parameter estimation (SSPE) using particle filter-based data assimilation is applied to predict displacement of the landslide.

Landslide SSPE assimilation strategy can make use of time-series displacements and hydrological information for the joint estimation of landslide displacement and model parameters, which can improve the performance considerably. We select Xishan Village, Sichuan province, China as experiment site to test SSPE assimilation strategy. Based on the comparison of actual monitoring data with prediction values, results strongly suggest the effectiveness and feasibility of SSPE assimilation strategy in short-term landslide displacement estimation.

## 1   Introduction

Landslide is a common geological hazard which greatly endangers the security of property and lives of the people (Huang et al., 2017). The landslide in Sri Lanka in May 2017 resulted in more than 200 people died and 698,289 people injuries (Kumarasiri, 2018). In China, the landslide hazard accounts for about 72.6% of the total geological disasters from 2005 to

2014(Xue et al., 2016). Therefore, landslide research is a hot topic studied by people, and it is necessary to do some prevention study like the early warning and deformation prediction (Liu et al., 2014; Jiang et al., 2016; Michoud et al., 2016).

Landslide prediction and forecast method had been developed and improved continually (Crosta et al., 2013; Li et al., 2018.). Chaussard E(2014) used the time-series analysis method of ALOS data to resolve land displacement in the Mexico region. Dong L (2012) proposed a model coupled Gray method and General Regression Neural Network (GM-GRNN) and

applied it to the prediction of sliding deformation of Dahu landslide. Li X Z (2014) carried out a genetic algorithm and support vector machine (GA–SVM) method to establish a mathematical function prediction model. However, most of the current model-based prediction cannot use the newest observation data effectively and deviate from the actual value. Data assimilation





method is a new technology that can help to deal with such problems. By combining surface observational data with the process model, data assimilation provides an optimal "true value" that is continuously distributed over time and space (Xue C et al. 2018). Data assimilation has been widely tested and used in geoscience fields like hydrologic and atmospheric and field(Reichle R H et al. 2002; Abbaszadeh P et al. 2017; Wikle C K et al. 2002). However, there are few explorations in the landslide field.

Data assimilation can be divided into two types: sequential-based method and continuous-based method (Qin J et al. 2009). A sequential-based way is an online approach that updates the prediction in each time (Nakano S. 2007), so it is more suitable for landslide system than continuous-based method. Particle Filter (PF) is a typical sequential data assimilation algorithm which was initially put forward by Gordon (Gordon N J et al. 2002). Because the PF is nonlinear filtering based on Bayesian estimation, it can deal with non-linear and non-Gaussian problems (Moradkhani H et al. 2011). Refer to the geosciences field. The numerical model shows large nonlinearity (Leeuwen P J V. 2010). So we choose PF as the algorithm to integrate multi-source data with the model.

The evolution of landslide is a time-varying process, so the model parameters are required to be adjusted over time. However, the primal sequential data assimilation only updates state vector, and the model parameters are generally given by known information, which will result in discrepancies between state and model parameters under a particular model relationship (Nearing G S et al., 2012). To meet the requirements of updating state values and model parameters simultaneously, we apply the simultaneous states and parameters estimation (SSPE) here. The SSPE method can continuously renewal the output by sequentially merging new measurements. Moradkhani (2005) optimized it into the hydrological field. Vrugt (2006) combined the simultaneous optimization with data assimilation. Joint estimation of state-parameter has proven to be a useful strategy to improve prediction performance (Qin J et al. 2009; Lü H et al. 2011).

In this paper, we applied the SSPE assimilation strategy to predict the landslide displacement. In landslide SSPE assimilation, the external factor hydrologic data has integrated into the dynamic model of landslide deformation data, which can adjust model parameters and state vector simultaneously according to the hydrologic information. During the process, internal factor of the landslide is combined with external observation factors. Thus reducing error and getting better predictive effects.

First, the research method will be deduced including time-series decomposition, establishment of the model and landslide SSPE assimilation strategy with PF algorithm. Then Xishan Village landslide is taken as the study area to examine SSPE assimilation strategy. The prediction of deformation will be optimized by coupling GPS observation data and hydrological factor. Finally, the study results will be proposed.





## 2 Method

### 2.1 Time-series displacement decomposition

Landslide deformation is the interaction between internal geological conditions and the external environment (Desai et al., 1995), so the displacement can be divided into the trend term displacement generated by inter factor, the periodic term

displacement caused by external factor (such as rainfall and reservoir water level, etc.) and the stochastic term displacement (human impacted, snowpack, etc.)(Zhou C et al., 2016). However, after noise filtering, the random term is too small and can be neglected. So the time-series displacement decomposition is as follows:

$$S(i) = \varphi(i) + x(i) \tag{1}$$

where $S(i)$ is the cumulative displacement of landslides, $\varphi(i)$ denotes the trend term, $x(i)$ denotes the periodic term.

The trend term of time series is extracted with moving average method because it can remove the disturbance effectively and leave long-term signals for research (Seng H. 2014).

$$\varphi_i = \frac{S_{i-1} + S_{i-2} + \cdots S_{i-n}}{n} \tag{2}$$

where $\varphi_i$ is the periodic term of step i, $S_{i-1}$ is the cumulative displacement of step i-1, n is the moving average period.

So the periodic term displacement can be calculated using subtraction between the total displacement and the trend term.

### 2.2 Landslide periodic displacement modeling

For the rainfall-induced landslide, atmospheric rainfall is one of the most susceptible disaster-causing factors and directly affects the periodic displacement of the landslide (Lian C et al., 2015; Ren F et al., 2015). So the periodic term can be regarded as a function of time and rainfall. The numerical function method is adopted here for the establishment of periodic displacement

model. The periodic displacement variation is tiny in a short time. Therefore, the model can be derived through expanding periodic displacement value using a Taylor-series expansion method:

$$x(t_{i+1}, r_{i+1}) = x(t_i, r_i) + (\frac{\partial x}{\partial t})_{t_i}(t_{i+1} - t_i) + \frac{1}{2}(\frac{\partial^2 x}{\partial t^2})_{t_i}(t_{i+1} - t_i)^2 + (\frac{\partial x}{\partial r})_{r_i}(r_{i+1} - r_i) + \frac{1}{2}(\frac{\partial^2 x}{\partial r^2})_{r_i}(r_{i+1} - r_i)^2 + g_i \tag{3}$$

where x denotes displacement of the landslide, $r_{i+1}$ is the rainfall of time i+1, $\frac{\partial x}{\partial t}$ and $\frac{\partial x}{\partial r}$ are the first order partial derivative of displacement, $\frac{\partial^2 x}{\partial t^2}$ and $\frac{\partial^2 x}{\partial r^2}$ are the second order partial derivative, $g_i$ is the remainder of Taylor's expansion.


### 2.3 Landslide SSPE assimilation strategy using PF

#### 2.3.1 State estimation

The general state-space model for a nonlinear dynamic system is defined to be:

State model: $x_{i+1} = f(x_i, u_i) + v_{i+1} \tag{4}$

Observation model: $y_{i+1} = g(x_{i+1}) + w_{i+1}$ (5)

where $x$ is the state vector and $y$ is the observation vector, $i$ is time step, $f$ and $g$ is nonlinear functions forecasting the state and observation, $u$ represents the model parameters, $v$ is model error and w is observation noise.

### 2.3.2 Landslide SSPE method

In sequential data assimilation, SSPE algorithm can be applied through the state augmentation method (Chen T, 2005). Consider the model in Eq. (4), the original state vector $x_i$ is now augmented with the parameters u(t) to be

$$\boldsymbol{X_i} = \begin{bmatrix} x_i \\ u_i \end{bmatrix} \quad (6)$$

Apply simultaneous state-parameter estimation method into practical landslide state model in Eq. (3), the extended state vector can be expressed as:

$$\boldsymbol{X_i} = \left[x(t_{i,}, r_i) \quad (\frac{\partial x}{\partial t})_{t_i} \quad (\frac{\partial^2 x}{\partial t^2})_{t_i} \quad (\frac{\partial x}{\partial r})_{r_i} \quad (\frac{\partial^2 x}{\partial r^2})_{r_i}\right]^T \quad (7)$$

And we set:

$$(\frac{\partial x}{\partial t})_{t_{i+1}} = (\frac{\partial x}{\partial t})_{t_i} + (\frac{\partial^2 x}{\partial t^2})_{t_i}(t_{i+1} - t_i) + m_i \quad (8)$$

$$(\frac{\partial^2 x}{\partial t^2})_{t_{i+1}} = (\frac{\partial^2 x}{\partial t^2})_{t_i} + n_i \quad (9)$$

$$(\frac{\partial x}{\partial r})_{r_{i+1}} = (\frac{\partial x}{\partial r})_{r_i} + (\frac{\partial^2 x}{\partial r^2})_{r_i}(r_{i+1} - r_i) + u_i \quad (10)$$

$$(\frac{\partial^2 x}{\partial r^2})_{r_{i+1}} = (\frac{\partial^2 x}{\partial r^2})_{r_i} + v_i \quad (11)$$

Where $m_i$、$n_i$、$u_i$、$v_i$ are noise.

So the next moment $\boldsymbol{X_{i+1}}$ is：

$$\boldsymbol{X_{i+1}} = \begin{bmatrix} x(t_{i+1,}, r_{i+1}) \\ (\frac{\partial x}{\partial t})_{t_{i+1}} \\ (\frac{\partial^2 x}{\partial t^2})_{t_{i+1}} \\ (\frac{\partial x}{\partial r})_{r_{i+1}} \\ (\frac{\partial^2 x}{\partial r^2})_{r_{i+1}} \end{bmatrix} =$$

$$\begin{bmatrix} x(t_{i,}, r_i) + (\frac{\partial x}{\partial t})_{t_i}(t_{i+1} - t_i) + \frac{1}{2}(\frac{\partial^2 x}{\partial t^2})_{t_i}(t_{i+1} - t_i)^2 + (\frac{\partial x}{\partial r})_{r_i}(r_{i+1} - r_i) + \frac{1}{2}(\frac{\partial^2 x}{\partial r^2})_{r_i}(r_{i+1} - r_i)^2 + g_i \\ (\frac{\partial x}{\partial t})_{t_i} + (\frac{\partial^2 x}{\partial t^2})_{t_i}(t_{i+1} - t_i) + m_i \\ (\frac{\partial^2 x}{\partial t^2})_{t_i} + n_i \\ (\frac{\partial x}{\partial r})_{r_i} + (\frac{\partial^2 x}{\partial r^2})_{r_i}(r_{i+1} - r_i) + u_i \\ (\frac{\partial^2 x}{\partial r^2})_{r_i} + v_i \end{bmatrix} =$$





$$\begin{bmatrix} 1 & t_{i+1}-t_i & \frac{1}{2}(t_{i+1}-t_i)^2 & r_{i+1}-r_i & \frac{1}{2}(r_{i+1}-r_i)^2 \\ 0 & 1 & t_{i+1}-t_i & 0 & 0 \\ 0 & 0 & 1 & 0 & 0 \\ 0 & 0 & 0 & 1 & r_{i+1}-r_i \\ 0 & 0 & 0 & 0 & 1 \end{bmatrix} \cdot \begin{bmatrix} x(t_i,r_i) \\ (\frac{\partial x}{\partial t})_{t_i} \\ (\frac{\partial^2 x}{\partial t^2})_{t_i} \\ (\frac{\partial x}{\partial r})_{r_i} \\ (\frac{\partial^2 x}{\partial r^2})_{r_i} \end{bmatrix} + \begin{bmatrix} g_i \\ m_i \\ n_i \\ u_i \\ v_i \end{bmatrix} =$$

$$\begin{bmatrix} 1 & t_{i+1}-t_i & \frac{1}{2}(t_{i+1}-t_i)^2 & r_{i+1}-r_i & \frac{1}{2}(r_{i+1}-r_i)^2 \\ 0 & 1 & t_{i+1}-t_i & 0 & 0 \\ 0 & 0 & 1 & 0 & 0 \\ 0 & 0 & 0 & 1 & r_{i+1}-r_i \\ 0 & 0 & 0 & 0 & 1 \end{bmatrix} \cdot X_i + \begin{bmatrix} g_i \\ m_i \\ n_i \\ u_i \\ v_i \end{bmatrix} \qquad (12)$$

In Eq. (16) we make $\begin{bmatrix} 1 & t_{i+1}-t_i & \frac{1}{2}(t_{i+1}-t_i)^2 & r_{i+1}-r_i & \frac{1}{2}(r_{i+1}-r_i)^2 \\ 0 & 1 & t_{i+1}-t_i & 0 & 0 \\ 0 & 0 & 1 & 0 & 0 \\ 0 & 0 & 0 & 1 & r_{i+1}-r_i \\ 0 & 0 & 0 & 0 & 1 \end{bmatrix} = \omega_{i+1}$, $\begin{bmatrix} g_i \\ m_i \\ n_i \\ u_i \\ v_i \end{bmatrix} = \varepsilon_{i+1}$, so Eq.(12) can be

expressed as:

$$X_{i+1} = \omega_{i+1} * X_i + \varepsilon_{i+1} \qquad (13)$$

Refer to observation of landslide deformation:

$$y_{i+1} = x_i + w_{i+1} \qquad (14)$$

Combined the two expressions Eq. (13) and Eq. (14), we can build landslide SSPE state-space model to joint estimate

the landslide periodic displacement and model parameters.

### 2.3.3 PF algorithm

However, some parameters in the landslide state space model Eq. (13) and Eq. (14) are difficult to obtain (e.g.

$(\frac{\partial x}{\partial t})_{t_i}$ , $(\frac{\partial^2 x}{\partial t^2})_{t_i}$). Based on Monte Carlo methods, PF can be adopted to solve this problem. Instead of calculating partial

derivative directly, PF generates a large number of samples (particles) to approximate the posterior probability of the states,

thus obtaining an optimal result (Maskell and Gordon, 2002).

From Bayesian theorem, the posterior probability of the states can be inferred through

(1) forecast:

$$p(x_i|y_{1:i-1}) = \int p(x_i|x_{i-1}) \ p(x_{i-1}|y_{1:i-1}) dx_{i-1} \qquad (15)$$

(2) update:

$$p(x_i|y_{1:i}) = \frac{p(y_i|x_i)p(x_i|y_{1:i-1})}{p(y_i|y_{1:i-1})} \qquad (16)$$

where i is time, $x_i$ is state vector, $y_i$ is observation vector, $y_{1:i} = \{y_1, y_2, \cdots, y_i\}$, $p(x_{i-1}|y_{1:i-1})$ is the posterior distribution

function (PDF) for time step i-1, $p(x_i|y_{1:i-1})$ is the prior distribution for time step i, $p(x_i|x_{i-1})$ can be derived from the



model.

In PF, the posterior probability of the states are approximated by discrete random measures defined by particles and a set of weights associated with particles:

$$\hat{p}(x_i|y_{1:i}) \approx \sum_{k=1}^{N} w_i^k \, \delta\left(x_{0:i} - x_{0:i}^k\right) \tag{17}$$

where $\hat{p}(x_k|y_{1:k})$ is the approximate value of $p(x_k|y_{1:k})$, $x_{0:i}^k$ $and$ $w_k^i$ are particles and associated weight and $\sum_{i=1}^{N} w_k^i = 1$, $\delta$ denotes Dirac delta function.

The direct sampling of target $p(x_k|y_{1:k})$ is difficult. The sequential importance sampling (SIS) is considered here to solve this problem. The SIS gathers particles from a known density function and updates the importance weights by using the iterative method (Doucet A. et al. 2000). Meanwhile, the sampling importance resampling (SIR) is used to avoid the particles

deviate away from the truth value (Gordon N J et al. 2002). The SIR algorithm accumulate particles in high importance weight. So the estimates of the state vector come into

$$\hat{x}_i = \sum_{k=1}^{N} x_i^k w_i^k \tag{18}$$

So, the procedure of landslide SSPE assimilation strategy is shown in Fig. 1.

## 3  Study area and data

### 3.1  Study area

The study area is located in Xishan Village, Li County, Sichuan Province, China (Fig. 2), and it is in the upper part of the left bank slope of the Zagunao River. This landslide is a massive accumulative landslide, which is composed of three slopes. The slope of this landslide is about 25° ~ 45°. The length is about 4200m, and the width is around 1700m. The altitude of the

leading edge is 1500m, and the trailing edge is 3400m. Thus the altitude difference is 1900m. The landform undulation leads to apparent local variations. Xishan Village landslide has a 52m thick sliding which can lead to about 85 million m³. Before 2008, many cracks appeared in the front and middle of this landslide, causing a direct economic loss of 0.5 million yuan and affecting 189 people. The creep deformation intensified after May 2008 Wenchuan earthquake which threatening the security of residents' lives and properties. According to estimates, the potential economic loss is 50 million yuan. In the purpose of

reducing the damage and giving the early warning, the study is taken to forecast the deformation of this landslide.

### 3.2  Data introduction

### 3.2.1 GPS derived time-series displacement

The Xishan Village landslide has been set up 5 GPS continuous observation stations for obtaining the deformation value.

The GPS receivers were connected to the network. So the observation data could be transferred in real time. At the same time, a GPS reference station was set in the stability area as the reference for calculation. Fig.3 shows the distribution of all stations. After the GPS baseline calculation, we get the deformation of every observation station from August 2015 to June 2017. Fig.4 shows the final results. Due to the transmission problem, there are several gaps in the data. The interpolation method is considered to be taken here to fill the missing data (Velicer and Colby, 2005; Lenda G et al., 2016).

### 3.2.2 Rainfall data

There are two rain gauges be settled in the landslide range of Xishan Village, which can transmit rainfall data in real time. The daily rainfall data is illustrated in Fig.5. Since the rain gauges are located near the GPS station, the mean values of the two gauges are taken as the rainfall of Xishan landslide.

### 4    Results and analysis

In this experiment, the SSPE assimilation method and model method without SSPE or data assimilation are used to predict landslide displacement. All the experimental data was obtained at Xishan Village landslide between August 2015 and June 2017. We only illustrate with two stations GPS03 and GPS04 because the deformation is more evident than others. The time step is set to five days with the purpose of saving computational time and increasing numerical accuracy correspondingly. The prediction work can be divided into the prediction of trend term, period term and cumulative term. Then the error analysis is taken to validate the efficiency of our method.

### 4.1    Prediction of trend term displacement

The trend term displacement is time monotone function so that it can be fitted by a polynomial, and the best results of trend term prediction and fitting formula are as shown in Fig. 6.

### 4.2    Prediction of period term displacement

The periodic term displacement can be calculated using the difference between the total displacement and the trend term. Fig. 7 shows the periodic displacement in station GPS03 and GPS04 and the rainfall data. It can be seen clearly that the period term is a complex nonlinear sequence series. Besides, fluctuation of the period term in two stations has the relatively same changing tendency, which is lagged behind that of rainfall. So we applied the SSPE assimilation method to predict it. The prediction results are as shown in Fig. 8. It can be seen that SSPE assimilation method get more close to the measured value





than the model method without assimilation.

### 4.3 Prediction of cumulative displacement

Finally, the predicted values of cumulative displacement can be obtained by summing up the predicted values of trend

and periodic displacement. The prediction results in each station are shown in Fig. 9. Additionally, some detailed prediction

data are enumerated (Table. 1 and Table.2). Then we calculate the difference between prediction and measured value and error

rate. Experimental results verify the feasibility of SSPE assimilation method.

### 4.4 Relative error analysis

In this section, more quantitative analysis is carried out to assess the performance of each method. Three criterions: Mean

Absolute Error (MAE), Mean Squared Error (MSE) and Root Mean Square Error (RMSE) were used to evaluate the prediction

effect. They can measure the deviation between the predicted value and the measured value, which calculated as

$$MAE = \frac{1}{N} * \sum_{i=1}^{N} |x_i - \hat{x}_i| \tag{19}$$

$$MSE = \frac{1}{N} * \sum_{i=1}^{N} (x_i - \hat{x}_i)^2 \tag{20}$$

$$RMSE = \sqrt{\frac{1}{N} * \sum_{i=1}^{N} (x_i - \hat{x}_i)^2} \tag{21}$$

Where $x_i$ is the measured value and $\hat{x}_i$ is the prediction value.

The results are as shown in Table. 3. According to the prediction evaluation indexes, SSPE assimilation method offers

better forecast effect than the model method. The MAE, MSE and RMSE value of SSPE assimilation method were 46.27%,

70.99% and 46.25% lower than those of Model method in GPS03 station, and 39.48%,61.75% and 38.15% lower in GPS04

station. The result suggests that the SSPE assimilation method has achieved great performance in landslide displacement

prediction.

### 5 Conclusion

Aims at coupling landslide deformation with external factor meanwhile increasing the accuracy of landslide prediction,

this paper presents a practical strategy on landslide displacement prediction. During the research, one of data assimilation

algorithm, the PF improved with SSPE method had been presented and applied with the reality. In real data experiment,

landslide deformation from GPS measuring was decomposed into trend term and period term firstly. The period term was

predicted with the hydrological factor in simultaneous estimation data assimilation, while the trend term was computed by





polynomial fitting.

Our results show that SSPE assimilation strategy has an excellent ability for landslide displacement prediction and can provide assistance in early risk assessment and landslide forecasting. However, further analysis and modification are needed in the future, such as more influence factor and complete model.

*Competing interests*. The authors declare that they have no conflict of interest.

*Acknowledgements:* This study was financially supported by the National Program on Key Basic Research Project of China (grant numbers: 2013CB733205).

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



**Figures**

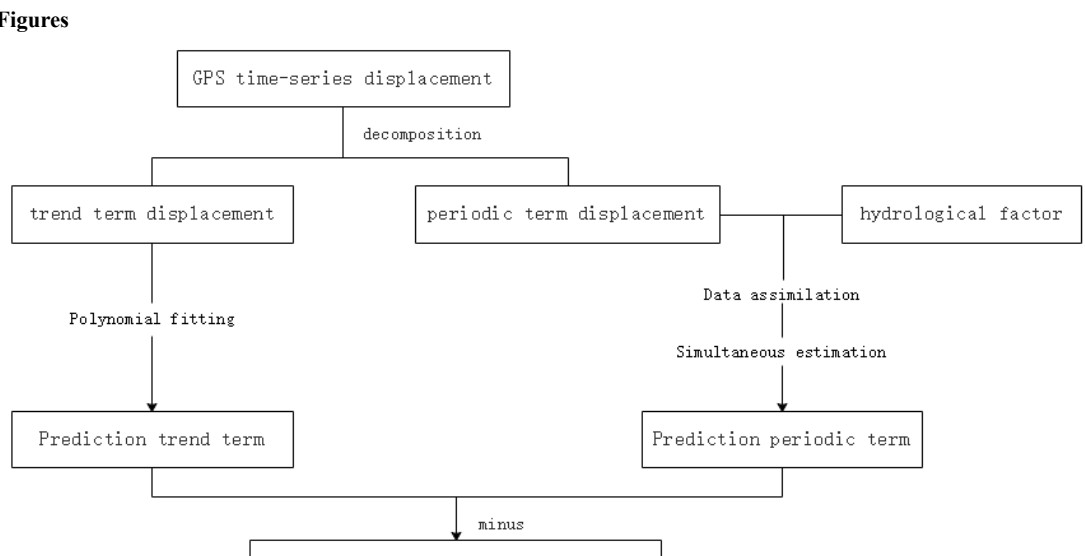

**Figure 1.** The flow chart of landslide SSPE assimilation strategy




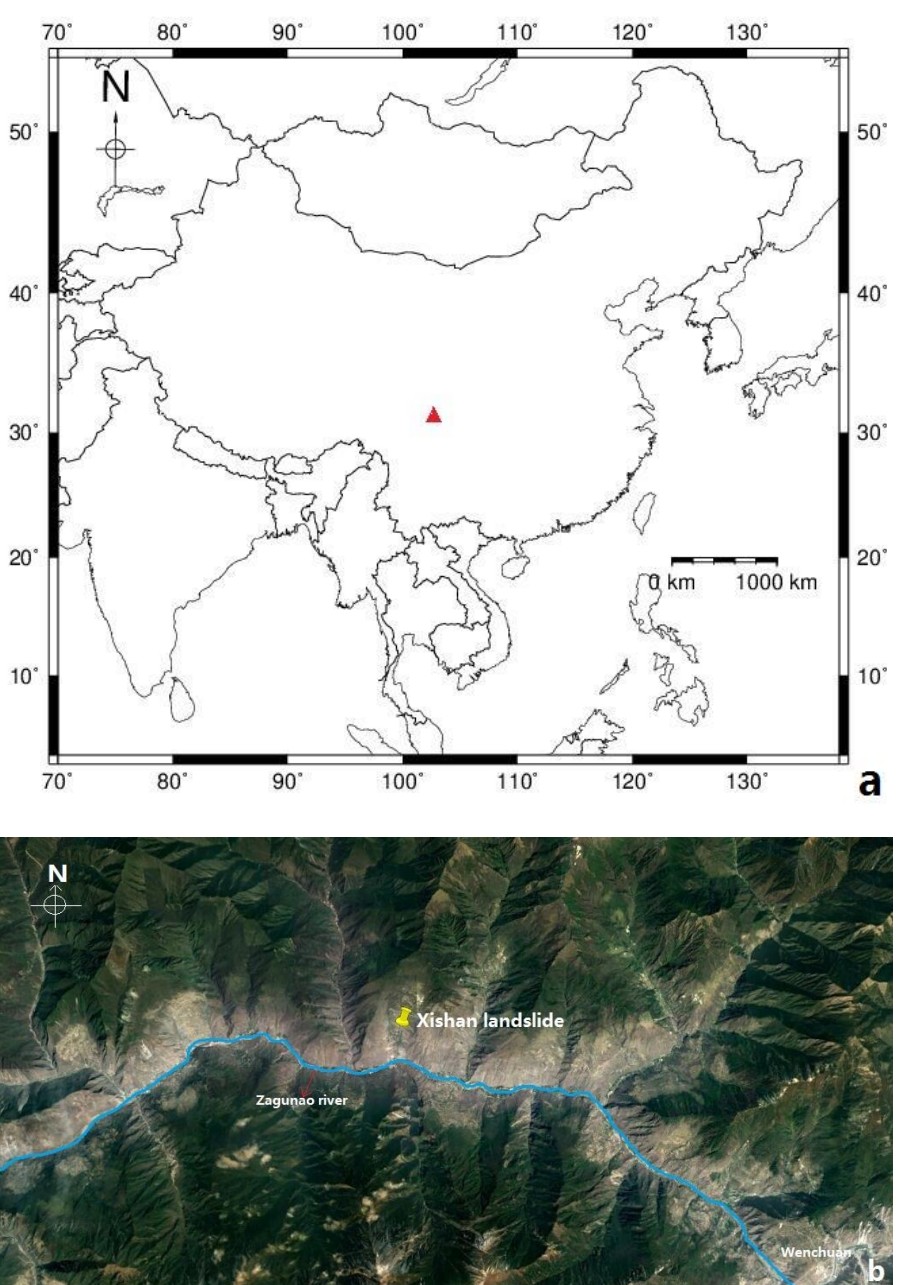

**Figure 2**. Location of the Xishan landslide in China (a); The Xishan landslides in the west of Wenchuan County associated with landform (b).



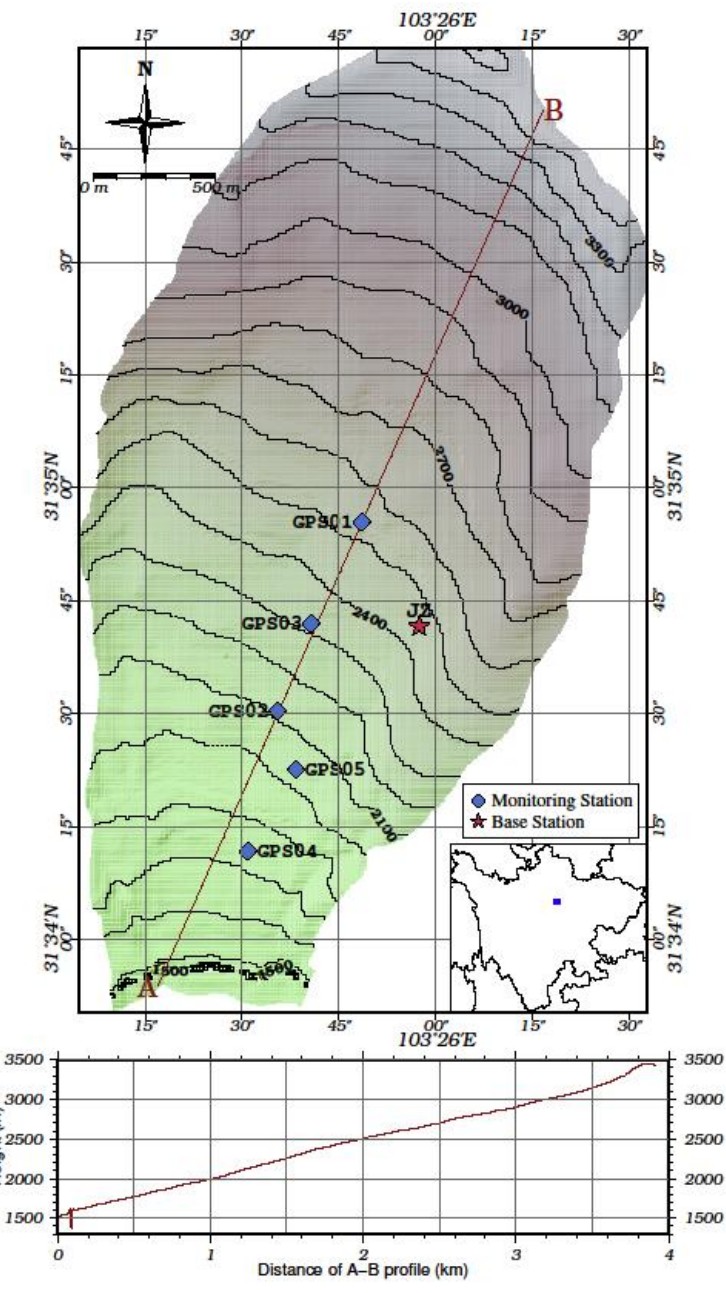

**Figure 3.** The distribution of GPS stations at Xishan landslide





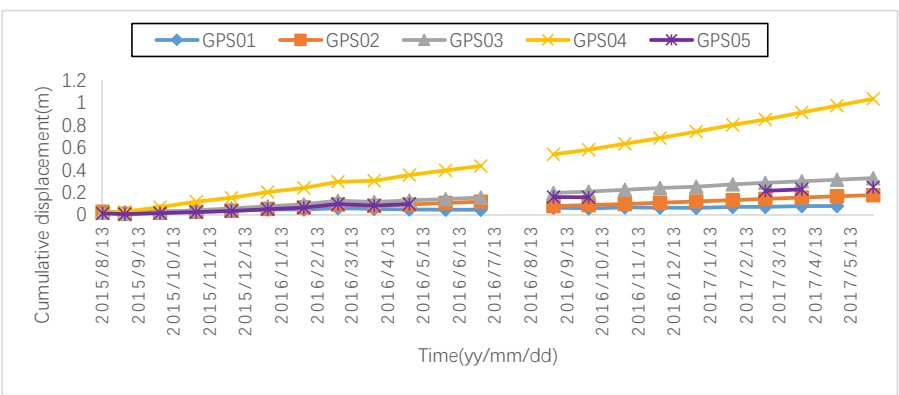

**Figure 4**. The GPS derived time-series displacement of Xishan landslide

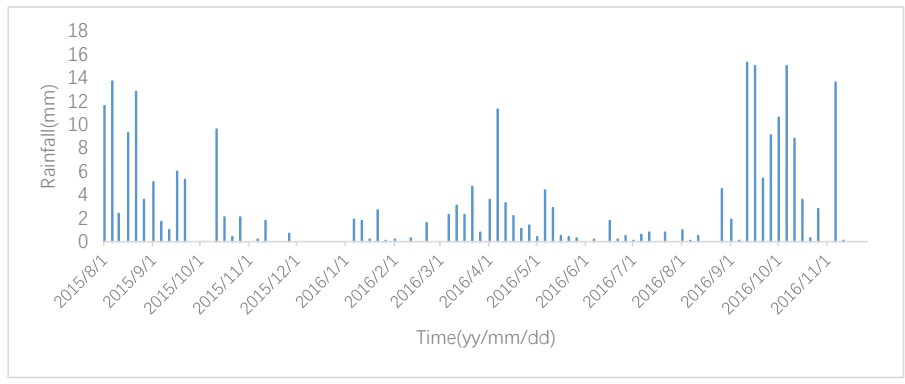

**Figure 5**. The rainfall of Xishan landslide



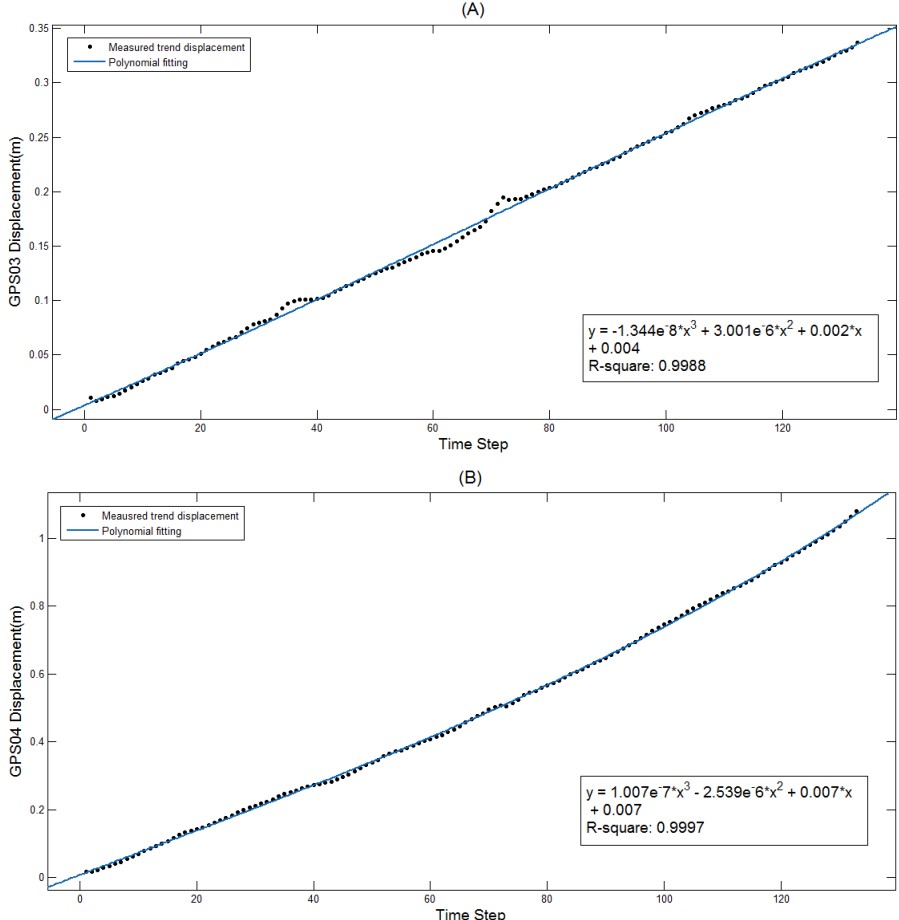

**Figure 6.** The trend term displacement prediction of (A)station GPS03, (B)station GPS04





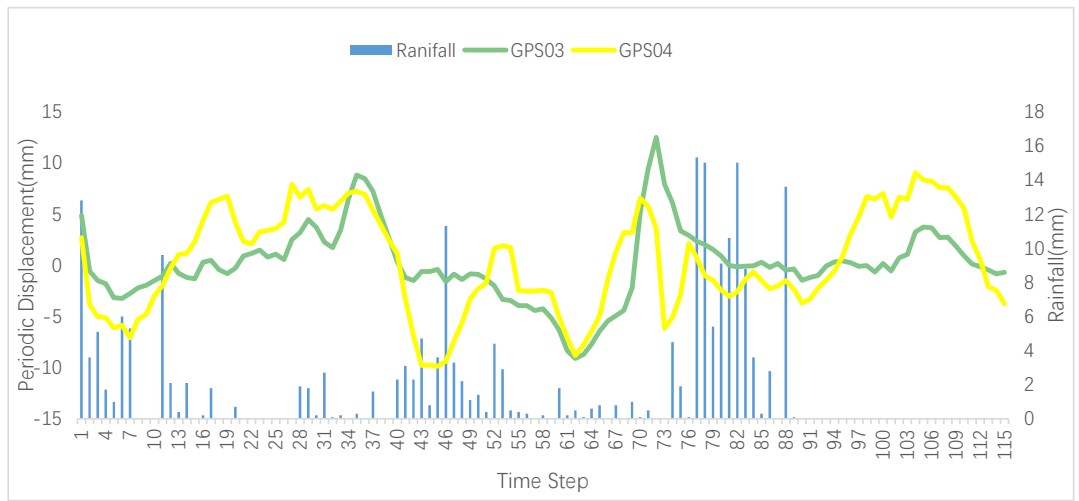

**Figure 7**. The periodic term displacement prediction combine with rainfall data in GPS03 and GPS04

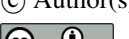


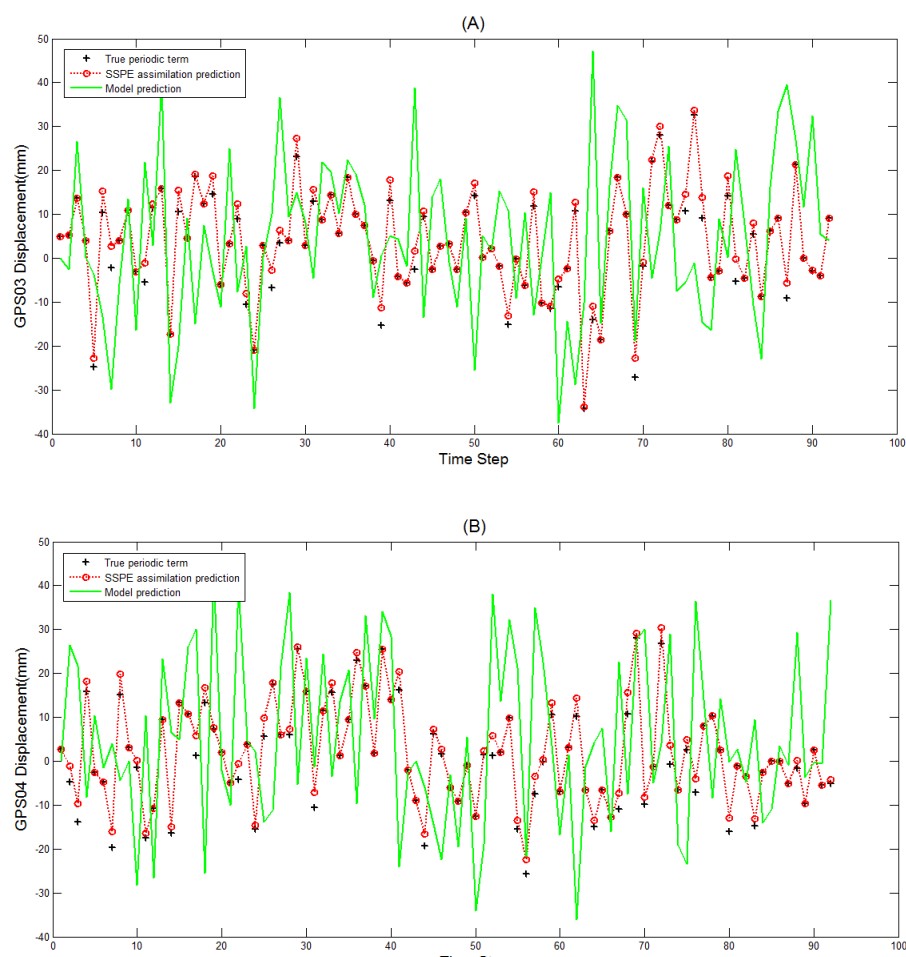

**Figure 8.** The periodic term displacement prediction of (A)station GPS03, (B)station GPS04




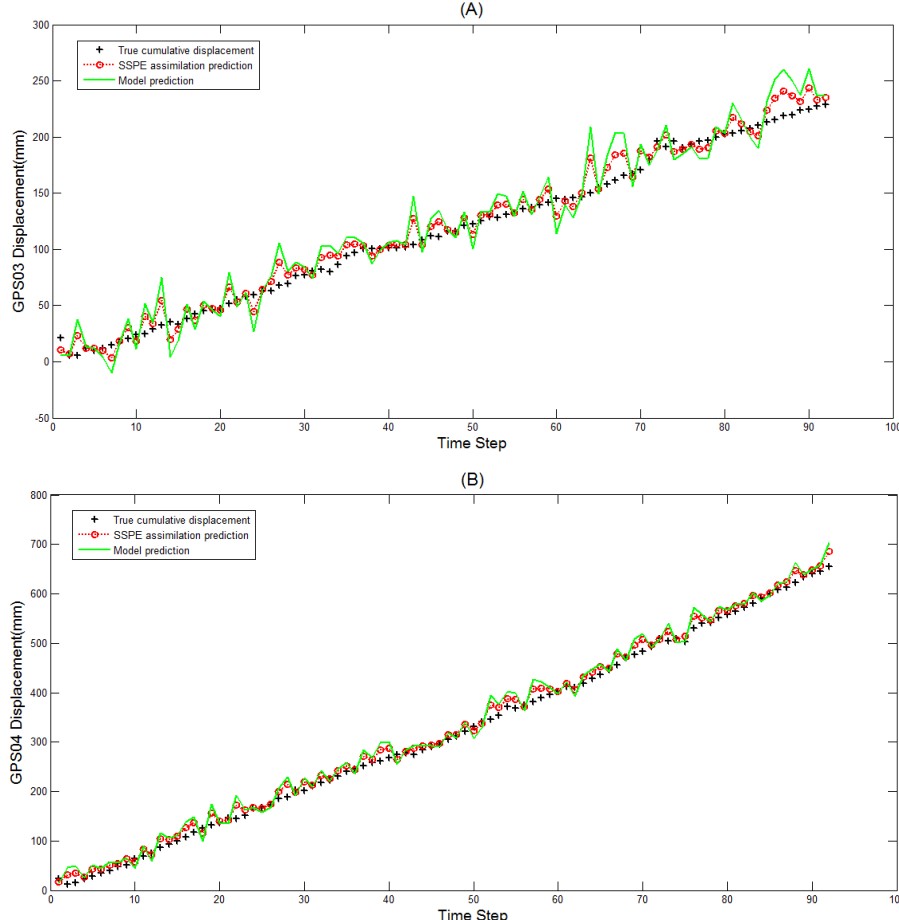

**Figure 9**. The cumulative displacement prediction of (A)station GPS03, (B)station GPS04



**Tables**

**Table 1** Comparison between the predicted values of cumulative displacement and measured displacement using different methods in station GPS03

| Time | Measured value (mm) | Model | | | SSPE assimilation | | |
|---|---|---|---|---|---|---|---|
| | | Prediction value (mm) | Difference (mm) | Error rate (%) | Prediction value (mm) | Difference (mm) | Error rate (%) |
| 2015/11/26 | 47.0829 | 40.3051 | 6.7778 | 14.40 | 45.7590 | 1.3239 | 2.81 |
| 2015/12/6 | 55.0449 | 48.7331 | 6.3118 | 11.47 | 53.1385 | 1.9064 | 3.46 |
| 2016/1/21 | 80.4837 | 74.0365 | 6.4472 | 8.01 | 77.4370 | 3.0467 | 3.79 |
| 2016/3/25 | 108.0314 | 97.6699 | 10.3615 | 9.59 | 104.0814 | 3.9501 | 3.66 |
| 2016/4/21 | 116.0395 | 110.1579 | 5.8816 | 5.07 | 114.9779 | 1.0616 | 0.91 |
| 2016/6/1 | 137.1018 | 131.1846 | 5.9173 | 4.32 | 135.4042 | 1.6976 | 1.24 |
| 2016/8/1 | 166.8250 | 155.9336 | 10.8915 | 6.53 | 164.1888 | 2.6363 | 1.58 |

**Table 2** Comparison between the predicted values of cumulative displacement and measured displacement using different methods in station GPS04

| Time | Measured value (mm) | Model | | | SSPE assimilation | | |
|---|---|---|---|---|---|---|---|
| | | Prediction value (mm) | Difference (mm) | Error rate (%) | Prediction value (mm) | Difference (mm) | Error rate (%) |
| 2015/10/21 | 75.7527 | 59.2630 | 16.4898 | 21.77 | 72.4569 | 3.2958 | 4.35 |
| 2015/12/1 | 146.0268 | 135.0044 | 11.0224 | 7.55 | 141.1449 | 4.8818 | 3.34 |
| 2016/1/11 | 203.0020 | 192.7036 | 10.2984 | 5.07 | 199.0452 | 3.9568 | 1.95 |
| 2016/3/11 | 274.8091 | 254.9907 | 19.8184 | 7.21 | 265.4005 | 9.4085 | 3.42 |
| 2016/4/26 | 331.1450 | 307.5421 | 23.6029 | 7.13 | 323.4631 | 7.6820 | 2.32 |
| 2016/5/26 | 374.9109 | 362.8496 | 12.0613 | 3.22 | 372.4185 | 2.4924 | 0.66 |
| 2016/8/21 | 504.4282 | 540.8718 | -36.4436 | -7.22 | 523.3166 | -18.8885 | -3.74 |

**Table 3.** Comparison of MAE, MSE, RMSE performance using different methods in two stations

| Method | MAE(mm) | | MSE(mm) | | RMSE(mm) | |
|---|---|---|---|---|---|---|
| | GPS03 | GPS04 | GPS03 | GPS04 | GPS03 | GPS04 |
| SSPE assimilation | 8.206 | 10.9272 | 110.9938 | 187.99 | 10.5137 | 13.7111 |
| Model | 15.2718 | 18.0565 · | 382.5577 | 491.46 | 19.5591 | 22.1689 |