# Peer review of "Simultaneous state-parameter estimation of rainfall-induced"

_Natural Hazards and Earth System Sciences, 2019_

## Referee Comment (RC1) · Anonymous Referee #1 · 10 Mar 2019

General comments: Prediction of landslide movement is essential for early warning against landslide disaster. It should be simultaneous especially in the case of rapid landslides for practical use of the prediction. This paper developed the simultaneous prediction method for landslide movements for the practical early warning particle by adopting particle filter-based data assimilation. Developing this kind of simultaneous prediction can contribute to practical early warning. Precision of data assimilation was described while merits of the proposed method, quickness of estimation of model parameter, could not be fully described according to some aspects in the paper as below.

Specific comments: (1) page 1, lines31-32: Please show calculating time needed for

[Figure]

existing prediction method. These examples can support superiority of prosed method in this paper. (2) page 3, lines 16 – 25: Geology in study area should be described. This is basic information for the landslide. (3) page 6, line 18: Show "three slopes" in the landslide in Figure 2 or Figure 3 and explain why the slope in Figure 3 was selected for the study. (4) page 7, lines 7 – 10: Show the location in Figure 2 or 3 and distance from the rain gauge to the monitored site. (5) page 7, line 13: Explain "model method without SSPE" in chapter 2. I could not find the model and method for deciding model parameters in chapter 2. If not, readers cannot the content of the paper. (6) page 7, lines 15 – 16: Please explain how to decide time step for calculation. It is important information for understanding the calculation process. (7) page 7, lines 27-28: Trend of fluctuation is different between GPS03 and GPS04. For examples, Fluctuation from time step 40 to 50 and that from 70 to 76. More detailed explanation is necessary for them. (8) page 8, lines 9 to 21: If you can show the comparison of calculating time needed for proposed method to that for the model without SSPE, readers can understand the superiority of the proposed method better. You can show the proposed method can make prediction simultaneously.

---

## Referee Comment (RC2) · Anonymous Referee #2 · 21 Mar 2019

The manuscript describes a method for predicting landslide displacement using simultaneous state-parameter estimation (SSPE) with data assimilation and shows application of the method to a landslide in Sichuan, China. The paper is concise and well organized. References are mostly adequate and current. The English is mostly adequate, but the paper would benefit from editing to improve clarity and grammar.

I have two main criticisms of this paper: (1) The polynomial trend line alone (Fig. 6) seems to be a better predictor of the net displacement than the full method (SSPE accumulation prediction + polynomial trend line, Fig. 9). (2) The authors have not shown conclusively that their proposed method works in a situation that is critical for
hazard management, namely, predicting when landslide displacements will accelerate to rapid, dangerous movement.

Regarding point 1, above, visual comparison of the different plots in Figures 6, 7, 8, and 9 is complicated by different vertical scales and lengths of the time series shown. Please show dates rather than time steps and use the same starting and ending dates for each of the plots in Figs. 6,7,8, and 9. Inconsistencies between amplitudes of observed periodic displacements in Fig. 7 (+/-10 mm) and Fig. 8 (+/-30 mm) cast doubt on results of the SSPE assimilation prediction in Figs. 8 and 9. Nevertheless, the overall displacement pattern (Fig. 6 and 9) is relatively steady (gradually changing) and the periodic component of displacement appears to account for only a tiny fraction of the net displacement. At the five-day time step used, the additional complication of predicting the periodic displacement component seems unnecessary for the data from stations GPS03 and GPS04.

Regarding point 2 above, relatively steady movement like that observed at stations GPS03 and GPS04 is relatively easy to predict and does not require a very sophisticated model to predict the displacement pattern, as discussed above. It is important that any method of predicting landslide displacement perform well in predicting relatively steady or gradually accelerating and decelerating movement and the time series from stations GPS03 and GS04 are well suited for such a test. An additional displacement time series that shows quick acceleration from slow movement to very rapid movement would provide an important practical test of the method. Displacement predictions in such a case would require shorter time steps (perhaps variable time steps that get shorter as displacement rate increases). How would that affect model performance (accuracy and computational speed?

Additional comments: Show an outline of the landslide boundaries on Fig. 3. Page 1, line 22, A more general reference about landslide danger to life and property is needed here. Page 2, lines 18 and 19, check citations against reference list.

---

## Author Comment (AC1) · 21 Mar 2019

Dear Referee:

Thank you very much for your good evaluation and kind comments concerning our manuscript entitled "Simultaneous state-parameter estimation of rainfall-induced landslide displacement using data assimilation". Those comments are valuable and helpful for revising and improving our paper. We have studied comments carefully and have made extensive modification on the original manuscript. A revised manuscript with the correction portion red marked is attached in the supplement and we hope meet with approval. The main corrections in the paper and the responds to the comments are as

follows:

Answers to comments:

(1) page 1, lines31-32: Please show calculating time needed for existing prediction method. These examples can support superiority of prosed method in this paper.

Response: Thanks for the advice. We have found our deficiencies in the work. We have reviewed many papers on landslide prediction and found that the detailed calculating time is not mentioned. We have tried our best to explain the shortcomings of existing method in page 1, line 31-31 and page 2, line 1-3;

(2) page 3, lines16-25: Geology in study area should be described. This is basic information for the landslide.

Response: Thanks for the kind suggestion. We have added the geology information of study area in page 6, line 24-29. The added section introduces the basic geologic feature and geologic structure of study area;

(3) page 6, line 18: Show "three slopes" in the landslide in Figure 2 or Figure 3 and explain why the slope in Figure 3 was selected for the study.

Response: We are very sorry for our inappropriate writing. It ought to be "three parts". All parts belong to our study area. We have artificially divided the total landslide into three slopes according to the geomophogensis. The necessary change in the paper has been made in page 6, line 24-27 as well as in the referred Figure 3 accordingly;

(4) page 7, lines 7 – 10: Show the location in Figure 2 or 3 and distance from the rain gauge to the monitored site.

Response: The location of rain gauges are illustrated in Figure 3. The distance between rain gauge and monitored site is less than a meter and expounded in page 7, line 19;

(5) page 7, line 13: Explain "model method without SSPE" in chapter 2. I could not

find the model and method for deciding model parameters in chapter 2. If not, readers cannot the content of the paper.

Response: We have revise this negligence in page 7, line 24-26. The "model method" is a method using SSPE strategy without data assimilation. And the calculation of parameters follows the formula Eq.(13) in page 5, line 6;

(6) page 7, lines 15 – 16: Please explain how to decide time step for calculation. It is important information for understanding the calculation process.

Response: The reason why we choose five days as time step is added in page 7, line 28-29. Due to the complex terrain and insufficient power supply of Xishan Village, the monitoring GPS sequence had large error or noise. In order to reduce the influence of these errors and noises, the time step is set to five days;

(7) page 7, lines 27-28: Trend of fluctuation is different between GPS03 and GPS04. For examples, Fluctuation from time step 40 to 50 and that from 70 to 76. More detailed explanation is necessary for them.

Response: The detailed explanation is as shown in page 8, line 12-16. The different fluctuation could be attributed to the impact of geology. The location of GPS 03 and 04 station have different geomophogensis, which will result in different deformation behavior;

(8) page 8, lines 9 to 21: If you can show the comparison of calculating time needed for proposed method to that for the model without SSPE, readers can understand the superiority of the proposed method better. You can show the proposed method can make prediction simultaneously.

Response: We have fixed our program and calculated the needed time of two algorithm. The model method predicts displacement without data assimilation algorithm, so it needs less time than SSPE method. The detailed data is shown in Table 3. The instructions are mentioned in page 9, line 9-11.

We are very grateful to your comments for the manuscript. Should you have any questions, please contact us without hesitate.

Please also note the supplement to this comment:
https://www.nat-hazards-earth-syst-sci-discuss.net/nhess-2019-24/nhess-2019-24-AC1-supplement.pdf
* * *
[Figure]

**Fig. 1.** Figure3:The distribution of three parts, GPS stations and rain gauges at Xishan landslide

| Method | MAE(mm) | | MSE(mm) | | RMSE(mm) | | Execution time |
|---|---|---|---|---|---|---|---|
| | GPS03 | GPS04 | GPS03 | GPS04 | GPS03 | GPS04 | |
| SSPE assimilation | 8.206 | 10.9272 | 110.9938 | 187.99 | 10.5137 | 13.7111 | 0.0889 |
| Model | 15.2718 | 18.0565 | 382.5577 | 491.46 | 19.5591 | 22.1689 | 0.0025 |

**Fig. 2.** Table3: Comparison of MAE, MSE, RMSE performance and needed time using different methods in two stations

**Supplement:**

**Simultaneous state-parameter estimation of rainfall-induced landslide displacement using data assimilation**

Jing Wang[1], Guigen Nie[1,2], Shengjun Gao[3],Changhu Xue[1]

[1] GNSS Research Center, Wuhan University, Wuhan, 430079, China

[2] Collaborative Innovation Center for Geospatial Information Technology, Wuhan, 430079,China

[3]Chinese Antarctic Center of Surveying and Mapping, Wuhan, 430079, China

*Corresponding to*: Guigen Nie (ggnie@whu.edu.cn)

**Abstract**. Landslide displacement prediction has great practical engineering significance to landslide stability evaluation and early warning. The evolution of landslide is a complex dynamic process, applying classical prediction method will result in significant error. Data assimilation method offers a new way to merge multi-source data with the model. However, data assimilation is still deficient in the ability to meet the demand of dynamic landslide system. In this paper, simultaneous state-parameter estimation (SSPE) using particle filter-based data assimilation is applied to predict displacement of the landslide. Landslide SSPE assimilation strategy can make use of time-series displacements and hydrological information for the joint estimation of landslide displacement and model parameters, which can improve the performance considerably. We select Xishan Village, Sichuan province, China as experiment site to test SSPE assimilation strategy. Based on the comparison of actual monitoring data with prediction values, results strongly suggest the effectiveness and feasibility of SSPE assimilation strategy in short-term landslide displacement estimation.

**1    Introduction**

Landslide is a common geological hazard which greatly endangers the security of property and lives of the people (Huang et al., 2017). The landslide in Sri Lanka in May 2017 resulted in more than 200 people died and 698,289 people injuries (Kumarasiri, 2018). In China, the landslide hazard accounts for about 72.6% of the total geological disasters from 2005 to 2014(Xue et al., 2016). Therefore, landslide research is a hot topic studied by people, and it is necessary to do some prevention study like the early warning and deformation prediction (Liu et al., 2014; Jiang et al., 2016; Michoud et al., 2016).

Landslide prediction and forecast method had been developed and improved continually (Crosta et al., 2013; Li et al., 2018.). Chaussard E(2014) used the time-series analysis method of ALOS data to resolve land displacement in the Mexico region. Dong L (2012) proposed a model coupled Gray method and General Regression Neural Network (GM-GRNN) and applied it to the prediction of sliding deformation of Dahu landslide. Li X Z (2014) carried out a genetic algorithm and support vector machine (GA–SVM) method to establish a mathematical function prediction model. Although the above methods have certain practicability in the prediction of landslides, there is still a problem that the forecast of rainfall-induced landslides

[revised manuscript text omitted]

---

## Author Comment (AC2) · 2 Apr 2019

Dear referee:

Thank you very much for the review and the valuable comments on our manuscript entitled "Simultaneous state-parameter estimation of rainfall-induced landslide displacement using data assimilation". We have carefully revised the manuscript and answered the questions according to the suggestions. A revised document with the correction portion red marked is attached in the supplement . Because of your suggestions, the revised article becomes better and readers can get more valuable information. We sincerely hope this manuscript will be finally accepted. Thanks again to the editors and

reviewers for their help.

Answers to comments:

1.The polynomial trend line alone (Fig. 6) seems to be a better predictor of the net displacement than the full method (SSPE accumulation prediction + polynomial trend line, Fig. 9); Please show dates rather than time steps and use the same starting and ending dates for each of the plots in Figs. 6,7,8, and 9; Inconsistencies between amplitudes of observed periodic displacements in Fig. 7 (+/-10 mm) and Fig. 8 (+/-30 mm) cast doubt on results of the SSPE assimilation prediction in Figs. 8 and 9; The additional complication of predicting the periodic displacement component seems unnecessary for the data from stations GPS03 and GPS04.

Response: Thank you very much for your advices. We are very sorry for our inexact expression. The displacement in Fig.6 is trend displacement, not the net displacement. The trend displacement is a part of net displacement and shows quasi-linear characteristics, so it fits well by polynomial. The reason why we don't use polynomial trend line to predict net displacement is that the net displacement contains periodic term. The periodic term shows violent fluctuations so that it cannot fit by polynomial; I have modified the time coordinates show in Figures 6, 7, 8 and 9 according to the requirements; Please excuse our negligence. Because the rainfall data is about triple of the periodic displacement. We scale up the periodic displacement to make sure them on a similar scale. Figure 8 shows the expanded period term displacement. Thank you for your remind and we re-run the data assimilation experiment again. Since the selection of particles in the assimilation step is random, the results in Figure 8, 9 and Table 1, 2, 3 are different from the previous ones; The goal of our research is aimed at predicting the displacement more quickly and accurately. The periodic displacement in the initial period of the landslide is really small. But as the landslide develops it will gradually increase. I also hope that I can get more data in the future to verify my method.

2.The authors have not shown conclusively that their proposed method works in a
situation that is critical for hazard management; Relatively steady movement like that observed at stations GPS03 and GPS04 is relatively easy to predict and does not require a very sophisticated model to predict the displacement pattern; Displacement predictions in such a case would require shorter time steps (perhaps variable time steps that get shorter as displacement rate increases). How would that affect model performance (accuracy and computational speed?

Response: Thank you for your criticisms. We are unable to provide a conclusively warning value because the situation of each landslide is different. The induced factors are not only internal factors but also external human activities and so on. And the external human activities are difficult to predict; We have proceeded simulation experiments with particle filtering. It can also perform very well in the case of violent fluctuations. We think the advantage of our method is that it does not require a lot of geological exploration work in the early stage, which can save a lot of time to build a relatively accurate model and use a small amount of observation data to get the prediction result. It can be seen a near real-time method; We choose five days as a time step here just because of data quality control. The selection of the time step does not affect the performance of particle filter.

3.Show an outline of the landslide boundaries on Fig. 3.

Response: In Figure 3, the colored part is the entire area of the landslide.

4.Page 1, line 22. A more general reference about landslide danger to life and property is needed here.

Response: Thank you for your advice. We have added more reference in Page 1, line 23.

5.Page 2, lines 18 and 19, check citations against reference list.

Response: We have corrected the citations in Page 2, line 22 and 23.

We would like to express our sincere thanks again to the reviewers for the constructive

and positive comments. Should you have any questions, please contact us without hesitate.

Please also note the supplement to this comment:
https://www.nat-hazards-earth-syst-sci-discuss.net/nhess-2019-24/nhess-2019-24-AC2-supplement.pdf
* * *
[Figure]

(A)

y = -1.344e⁻8*x³ + 3.001e⁻6*x² + 0.002*x
+ 0.004
R-square: 0.9988

$$y = -1.344e^{-8}*x^3 + 3.001e^{-6}*x^2 + 0.002*x + 0.004$$
R-square: 0.9988

**Fig. 1.** Figure 6:The trend term displacement prediction of (A)station GPS03

(B)

The trend displacement plot for GPS04 station. X-axis: Time(yy/mm/dd) ranging from 2015/8/21 to 2017/4/16. Y-axis: GPS04 Trend Displacement(m) ranging from 0 to 1. Legend shows "Trend displacement" (dots) and "Polynomial fitting" (line).

$$y = 1.007e^{-7}*x^3 - 2.539e^{-6}*x^2 + 0.007*x + 0.007$$
R-square: 0.9997

**Fig. 2.** Figure 6:The trend term displacement prediction of (B)station GPS04

**Fig. 3.** Figure 7. The periodic term displacement combined with rainfall data in GPS03 and GPS04

(A)

Fig. 4. Figure 8. The periodic term displacement prediction of (A)station GPS03

[Figure]

Fig. 5. Figure 8. The periodic term displacement prediction of (B)station GPS04

[Figure]

**Fig. 6.** Figure 9. The cumulative displacement prediction of (A)station GPS03

(B)

GPS04 cumulative displacement(mm)

+ True cumulative displacement
····⊙···· SSPE assimilation prediction
—— SSPE prediction

Time(yy/mm/dd)

**Fig. 7.** Figure 9. The cumulative displacement prediction of (B)station GPS04

**Table 1** Comparison between the predicted values of cumulative displacement and measured displacement using different methods in station GPS03

| Time (yy/mm/dd) | Measured value (mm) | SSPE | | | SSPE assimilation | | |
|---|---|---|---|---|---|---|---|
| | | Prediction value (mm) | Difference (mm) | Error rate (%) | Prediction value (mm) | Difference (mm) | Error rate (%) |
| 2015/10/11 | 32.2674 | 40.2287 | -7.9614 | -24.67 | 29.1589 | 3.1085 | 9.63 |
| 2015/12/16 | 63.3499 | 68.1207 | -4.7708 | -7.53 | 61.8590 | 1.4909 | 2.35 |
| 2016/4/6 | 116.0395 | 105.4518 | 10.5878 | 9.12 | 115.0090 | 1.0305 | 0.89 |
| 2016/6/11 | 144.7729 | 133.5143 | 11.2586 | 7.78 | 145.9559 | -1.1830 | -0.82 |
| 2016/7/6 | 157.6520 | 146.3509 | 11.3011 | 7.16 | 156.2981 | 1.3539 | 0.86 |
| 2016/8/11 | 191.482 | 180.9944 | 10.4876 | 5.48 | 190.1751 | 1.3069 | 0.68 |
| 2016/10/16 | 215.3067 | 224.5674 | -9.2607 | -4.30 | 215.4657 | -0.1590 | -0.07 |
| 2016/11/21 | 233.1672 | 220.3506 | 12.8166 | 5.49 | 231.5734 | 1.5938 | 0.68 |

**Fig. 8.** Table 1 Comparison between the predicted values of cumulative displacement and measured displacement using different methods in station GPS03

[revised manuscript text omitted]

---

## Author Response (AR1)

Dear Editor Albert:

Thank you for your kind suggestion and detailed comments on our manuscript entitled "Simultaneous state-parameter estimation of rainfall-induced landslide displacement using data assimilation" (nhess-2019-24). We have carefully revised the manuscript according to the comments. All the revisions of the article are marked in red. We greatly appreciate your help concerning improvement of this manuscript.

**Answers to comments:**

1. So the manuscript most likely would benefit if you consider approaching an English support service to work on it as well.

   **Answers:** Thank you for your advice. Because of the rush of time, we don't have enough time to approach an English support service this time. We will follow your advice next time.

2. The rain data provided in figure 5 is very similar to that provided in figure 7, only the time span might have changed. If so, please remove figure 5 and adjust the references to this figure in manuscript where needed.

   **Answers:** Thank you for your detailed work. We have removed figure 5 and adjust the references in our manuscript.

We sincerely hope this manuscript will be finally accepted. Thanks again for your help.

With best wishes,

Yours sincerely,

Nie

The main changes in "nhess-2019-24-manuscript-version2.pdf (Date: 27 Apr 2019)":

1. Fix some grammar problem
2. Add more reference in Page 1, Line 23 and Page 2, Line 2-3
3. Add the geology information of study area in Page 6, line 24-28
4. Explanation the fluctuation in Page 8, Line10-14
5. Add more acknowledgements in Page9, Line 24-25
6. Add the parts and rain gauges in Figure 3
7. Remove the Figure 5 in version1 and fix the related reference
8. Modified the time coordinates show in Figures 5, 6, 7 and 8
9. Change the results in Figure 7, 8 and Table 1, 2, 3
10. Add the time of algorithm in Table 3

---

## Editor Decision (ED1)

[revised manuscript text omitted]
 geomorphology: The dip direction is about 178°; (Ⅱ) erosional and denudational geomorphology: The dip direction is about 200°; (Ⅲ)Glacial and periglacial geomorphology: The dip direction is about 208°. The distribution of three parts are as shown in Fig. 3. Xishan landslide is a soft rock slope of layered structure. According to the drilling, the exposed strata in the study area is mainly blue grey phyllite. And the upper deposit is collapse slope product and ice water accumulation, which is mainly composed of silt and gravel soil.

The landform undulation leads to apparent local variations. Xishan Village landslide has a 52m thick sliding which can lead to about 85 million m³. Before 2008, many cracks appeared in the front and middle of this landslide, causing a direct economic loss of 0.5 million yuan and affecting 189 people. The creep deformation intensified after May 2008 Wenchuan earthquake which threatening the security of residents' lives and properties. According to estimates, the potential economic loss is 50 million yuan. In the purpose of reducing the damage and giving the early warning, the study is taken to forecast the deformation of this landslide.

**3.2 Data introduction**

**3.2.1 GPS derived time-series displacement**

The Xishan Village landslide has been set up 5 GPS continuous observation stations for obtaining the deformation value. The GPS receivers were connected to the network. So the observation data could be transferred in real time. At the same time, a GPS reference station was set in the stability area as the reference for calculation. Fig.3 shows the distribution of all stations. After the GPS baseline calculation, we get the deformation of every observation station from August 2015 to June 2017. Fig.4 shows the final results. Due to the transmission problem, there are several gaps in the data. The interpolation method is considered to be taken here to fill the missing data (Velicer and Colby, 2005; Lenda G et al., 2016).

**3.2.2 Rainfall data**

There are two rain gauges be settled in the landslide range of Xishan Village, which can transmit rainfall data in real time. Fig.3 shows the location of rain gauges. Actually they are near to the GPS stations no more than a meter away. 
[revised manuscript text omitted]

---

## Author Response (AR2)

Dear Editor Albert Kettner:

Thank you for your comments concerning our manuscript entitled "Simultaneous state-parameter estimation of rainfall-induced landslide displacement using data assimilation" (nhess-2019-24). We have studied the comments carefully and have made correction which we hope meet with approval. All the revisions of the article are marked in red. We should like to express our appreciation to you for suggesting how to improve our paper.

**Answers to comments:**

1. Throughout the document, make sure that the references are cited properly.
   **Answers:** Thank you for your advice. We have corrected the citations in Page 1 Line 23, 29, 30, 31. Page 2 Line 2, 3, 7, 8, 9, 11, 12, 14, 15, 16, 21, 23, 24, 25. Page 3 Line 11, 16, 23. Page 4 Line 10. Page 6 Line 13, 14. Page 7 Line 15.

2. Page 1, line 13. Do the authors mean "Simultaneous state-parameter estimation" or as indicated in page 1, line "simultaneous states and parameters estimation"?
   **Answers:** We mean "simultaneous states and parameters estimation". And we have corrected this in Page 1 Line 13 and 14.

3. Page 5, line 9. "Combining the two ….. we can build the landslide …. and the model parameters."
   **Answers:** We have corrected this incorrect grammar in Page 5 Line 9 and 10.

4. Page 6, line 11. Remove 'is': "…is can be …"
   **Answers:** We have removed 'is' in Page 6, Line 11.

5. Page 6, line 21. 'of the left bank slope…'. Rephrase this such that it reads ' of the northern bank slope…'.
   **Answers:** We have rephrased it into "…northern bank slope…" in Page 6 Line 21.

6. Page 7, line 1. Rephrase so it reads "….can lead to the movement of about 85…."
   **Answers:** We have rephrased it into "…lead to the movement of about…" in Page 7 Line 2.

7. Page 7, line 9. Rephrase so it reads: "Five continuous GPS observation stations have been set up for the Xishan Village landslide for obtaining any deformation observations."
   **Answers:** We have rephrased it into "Five continuous GPS observation stations have been set up for the Xishan Village landslide for obtaining any deformation observations." in Page 7 Line 10 and 11.

8. Page 7, line 27. The monitoring GPS sequence contained significant noise or errors. In order to reduce the influence of this, the…
   **Answers:** We have rephrased it into "the monitoring GPS sequence contained significant noise or errors. In order to reduce the influence of this …" in Page 7 Line 28.

9. Page 7, line 27. Indicate how it was decided to take a 5day time step. Why not 3 days or 7 days? What procedure was used to find the optimum of 5 days?
   **Answers:** After experiment and evaluation among different time steps, five days' time step gives best correlation with rainfall data. Therefore the time step is set to five days. And we have added this in Page 7 Line 28 and 29 and Page 8 Line 1.

10. Page 8, line 9. Rephrase so it reads: "the period term of the two stations, both show relatively the same changing tendency, which….."
    **Answers:** We have rephrased it into "…the period term of the two stations, both show relatively the same changing tendency, which..." in Page 8 Line 11 and 12.

11. Page 8, line 11. Change text such that "The GPS04 station belongs to part | " reads "The GPS04

stations monitors the first part of the landslide".

**Answers:** We have rephrased it into "The GPS04 stations monitors the first part of the landslide." in Page 8 Line 13 and 14.

12. Page 8, line 12. There are a large number of people living there.

    **Answers:** We have rephrased it into "a large number of" in Page 8 Line 14.

13. Page 8, line 13. Change text such that "The GPS03 station belongs to part III." Reads "Station GPS03 monitors part III, the upper part of the landslide (Fig. 3)."

    **Answers:** We have rephrased it into "Station GPS03 monitors part III, the upper part of the landslide (Fig. 3)." in Page 8 Line 15 and 16.

14. Page 9, line 4. Add "a" so it reads "…. method offers a better forecast…."

    **Answers:** We have added "a" in Page 9 Line 6.

15. Page 9, line 8. "execution time of the two methods…."

    **Answers:** We have rephrased it into "…the two methods…" in Page 9 Line 10.

16. Page 9, line 24-25. The reference to google earth to figure 2 and rewrite to something like: "Map obtained from Google Earth".

    **Answers:** We have added "Map obtained from Google Earth" in Page 14 Figure 2.

17. Page 21, table 3. Write out the abbreviations so the table can be understood without reading the paper. So: "Comparison of Mean Absolute Error (MAE), Mean Squared Error (MSE) and Root Mean Square Error (RMSE) performance…."

    **Answers:** We have rephrased it into "Mean Absolute Error (MAE), Mean Squared Error (MSE) and Root Mean Square Error (RMSE)" in Page 21 Table3.

We greatly appreciate your help concerning improvement to this paper. We sincerely hope this manuscript will be finally accepted. Thanks again for your help.

With best wishes,

Yours sincerely,

Nie